# Diffusion Assessment of Cortical Changes, Induced by Traumatic Spinal Cord Injury

**DOI:** 10.3390/brainsci7020021

**Published:** 2017-02-17

**Authors:** Peng Sun, Rory K. J. Murphy, Paul Gamble, Ajit George, Sheng-Kwei Song, Wilson Z. Ray

**Affiliations:** 1Department of Radiology, Washington University School of Medicine, St. Louis, MO 63110, USA; pengsun@wustl.edu (P.S.); ajitgeorge@wustl.edu (A.G.); ssong@wustl.edu (S.-K.S.); 2Department of Neurological Surgery, Washington University School of Medicine, St. Louis, MO 63110, USA; murphyr@wudosis.wustl.edu; 3School of Medicine, Washington University, St. Louis, MO 63110, USA; gamblep@wusm.wustl.edu

**Keywords:** magnetic resonance imaging, diffusion tensor imaging, diffusion basis spectrum imaging, spinal cord injury

## Abstract

Promising treatments are being developed to promote functional recovery after spinal cord injury (SCI). Magnetic resonance imaging, specifically Diffusion Tensor Imaging (DTI) has been shown to non-invasively measure both axonal and myelin integrity following traumatic brain and SCI. A novel data-driven model-selection algorithm known as Diffusion Basis Spectrum Imaging (DBSI) has been proposed to more accurately delineate white matter injury. The objective of this study was to investigate whether DTI/DBSI changes that extend to level of the cerebral peduncle and internal capsule following a SCI could be correlated with clinical function. A prospective non-randomized cohort of 23 patients with chronic spinal cord injuries and 17 control subjects underwent cranial diffusion weighted imaging, followed by whole brain DTI and DBSI computations. Region-based analyses were performed on cerebral peduncle and internal capsule. Three subgroups of patients were included in the region-based analysis. Tract-Based Spatial Statistics (TBSS) was also applied to allow whole-brain white matter analysis between controls and all patients. Functional assessments were made using International Standards for Neurological Classification of Spinal Cord Injury (ISNCSCI) as modified by the American Spinal Injury Association (ASIA) Scale. Whole brain white matter analysis using TBSS finds no statistical difference between controls and all patients. Only cervical ASIA A/B patients in cerebral peduncle showed differences from controls in DTI and DBSI results with region-based analysis. Cervical ASIA A/B SCI patients had higher levels of axonal injury and edema/tissue loss as measured by DBSI at the level of the cerebral peduncle. DTI Fractional Anisotropy (FA), Axial Diffusivity (AD) and Radial Diffusivity (RD) was able to detect differences in cervical ASIA A/B patients, but were non-specific to pathologies. Increased water fraction indicated by DBSI non-restricted isotropic diffusion fraction in the cerebral peduncle, explains the simultaneously increased DTI AD and DTI RD values. Our results further demonstrate the utility of DTI to detect disruption in axonal integrity in white matter, yet a clear shortcoming in differentiating true axonal injury from inflammation/tissue loss. Our results suggest a preservation of axonal integrity at the cortical level and has implications for future regenerative clinical trials.

## 1. Introduction

Spinal cord injury (SCI) is a significant public health problem. Currently, 253,000 people in the United States are living with SCI, while 11,000 Americans are hospitalized for SCI each year [1]. Annually, $9.7 billion dollars are being spent on SCI research and patient care. A major shortcoming limiting efforts to improve the treatment of SCI is the lack of quantifiable metrics on which to base clinical decisions. Biomarkers are emerging in many fields as valuable predictors of a patient’s clinical course and response to therapy. Diffusion tensor imaging (DTI) has been demonstrated to noninvasively reflect the progression of white matter tract damage in SCI through characterizing water molecule diffusion [2,3,4,5,6,7,8]. The white matter of the central nervous system is highly ordered and has a coherent structure in which water diffusivity parallel to the fibers (axial diffusivity—AD) is greater than the diffusivity perpendicular to the fibers (radial diffusivity—RD). Changes in these directional diffusivities reflect white matter integrity and the underlying pathology [9,10,11,12]. Specifically, demyelination results in an increased RD, presumably due to the loss or disruption of myelin membrane integrity that hinders water diffusion perpendicular to axonal tracts [11,13,14]. In contrast, axonal injury leads to decreased AD levels [10,11,15].

In acute SCI, axonal loss, demyelination, edema, and cellular inflammation are all prevalent; and contribute to observed functional disability. In chronic SCI, cellular inflammation and edema are assumed to be less relevant, while axon and myelin loss are thought to be the primary substrate of functional disability. The neurological impairment of SCI patients may appear clinically similar while the underlying pathological profile may differ. Early axonal injury potentially leads to permanent disability, and may result in neurological impairment comparable with that from inflammation with minimal axonal damage, potentially reversible when inflammation or edema subsides. Early neurological impairment alone is thus not sufficient to predict the long-term outcome of a SCI patient.

While providing a valuable tool to assess white matter integrity, unfortunately it is now recognized that DTI loses specificity and sensitivity with increasing pathological and anatomical complexity [4,16,17]. Thus the prediction of long-term outcome using DTI remains uncertain. To overcome factors confounding DTI analysis, we have utilized diffusion basis spectrum imaging (DBSI) [4], to more accurately differentiate and quantify axonal injury, demyelination, inflammation and edema/tissue loss. While an increasing volume of work has been done utilizing DTI in traumatic brain injury, SCI, and cervical spondylotic myelopathy (CSM) it remains unclear whether brainstem and higher white matter tracts represent a useful predictor of clinical function following SCI. Furthermore, it is unclear what—if any—permanent cortical axonal loss occurs following a traumatic SCI. Recent work has demonstrated at least at the level of the spinal cord, FA values rostral to the site of injury correlate with clinical function in acute SCI [18]. We hypothesized DTI and DBSI metrics of the brainstem and internal capsule would also represent a useful predictor of clinical function in chronic SCI patients.

## 2. Materials and Methods

The Washington University Human Research Protection Office/Institutional Review Board and the Saint Louis University Institutional Review Board approved this cross-sectional study, and all subjects provided written informed consent. A prospective non-randomized cohort of 23 SCI patients and 17 control subjects underwent Magnetic Resonance Imaging (MRI) over a 12-month period. DTI and DBSI analyses were performed on whole brain diffusion weighted images. DTI and DBSI-derived indices from the internal capsule (IC) and cerebral peduncle (CP) were obtained. Clinical grading of patient functional status was made with international standards for neurological classification of spinal cord injury (ISNCSCI) [19] as modified by the American Spinal Injury Association (ASIA) Scale.

### 2.1. DTI/DBSI Image Acquisition

Magnetic resonance images were acquired using a 3 T scanner (Trio, Siemens, Erlangen, Germany) using a single-shot diffusion weighted Spin-Echo Echo Planar Imaging (SE-EPI) sequence with the following parameters: Repetition Time (TR) = 10,000 ms; Echo Time (TE) = 120 ms; Field of View (FOV) = 256 × 256 mm^2^; acquisition matrix = 128 × 128; slice thickness = 2 mm; in-plane resolution = 2 × 2 × 2 mm^3^; acquisition time = 15 min. Diffusion weighted images were acquired in the axial plane covering the whole brain. As listed in Table 4 of Appendix A, a total 99 diffusion-encoding directions were selected as prescribed in diffusion spectrum imaging where the position vectors are the entire grid points (*qx*, *qy*, *qz*) over the 3D q-space under the relationship that (*qx*^2^ + *qy*^2^ + *qz*^2^) ≤ *r*^2^ (*r* = 3 used in current study). There were nine distinct b values distributed from 0 to 2000 s/mm^2^ on the uniformly spaced Cartesian grid (0/200/450/650/900/1100/1350/1800/2000 s/mm^2^) [20,21,22]. Eddy current and motion artifacts of Diffusion Weight Image (DWI) images were corrected, then susceptibility-induced off-resonance field was estimated and corrected using TOPUP in FMRIB Software Library (FSL) [23]. Images from all subjects were registered to standard JHU-DTI-MNI single-subject atlas (also known as the “Eve Atlas” [24]). DTI and DBSI post-processing were conducted using software developed at the Mallinckrodt Institute of Radiology of Washington University in St. Louis. Standard Regions of Interest (ROIs) of CP and IC for each participant were selected based on the Johns Hopkins University (JHU) MNI template White Matter Parcellation Map (WMPM). Student’s *t*-test was used for statistical analysis of group difference.

### 2.2. Diffusion Basis Spectrum Imaging (DBSI)

DBSI [4] analyzed the diffusion MRI signals as a linear combination of multiple anisotropic (representing crossing myelinated and unmyelinated axons of varied directions; the first term) and a spectrum of isotropic (resulting from cells, sub-cellular structure, and edematous water; the second term) diffusion tensors according to Equation (1):
(1)$$S_{k}=\overset{N_{Aniso}}{\underset{i=1}{\sum }}f_{i}e^{-\left|\vec{b_{k}}\right|\lambda_{⊥i}}e^{-\left|\vec{b_{k}}\right|\left(\lambda_{∥i}-\lambda_{⊥i}\right)\cos^{2}\psi_{ik}}+\int_{a}^{b}f\left(D\right)e^{-\left|\vec{b_{k}}\right|D}dD\;\left(k=1,\;2,\;3,\ldots \right)\;$$
where $S_{k}$ and $\left|\vec{b_{k}}\right|$ are the signal and *b*-value of the *k^th^* diffusion gradient, $N_{Aniso}$ is the number of anisotropic tensors (fiber tracts), $\Psi_{ik}$ is the angle between the *k^th^* diffusion gradient and the principal direction of the *i^th^* anisotropic tensor, $\lambda_{||i}$ and $\lambda_{⊥i}$ are the axial and radial diffusivities of the *i^th^* anisotropic tensor, $f_{i}$ is the signal intensity fraction for the *i^th^* anisotropic tensor, and *a* and *b* are the low and high diffusivity limits for the isotropic diffusion spectrum (reflecting cellularity and edema) f(D) [4]. Since all pathology markers are derived using a single diffusion-weighted MRI data set in DBSI, naturally co-registered, the inherently quantitative relationship among all DBSI metrics enables the quantification of each pathological component.

### 2.3. Tract-Based Spatial Statistics (TBSS)

The whole brain voxel-wise analysis of DTI and DBSI were carried out using Tract Based Spatial Statistics (TBSS) [25], part of FSL [26]. Due to the limited number of patients, TBSS was only applied between controls and all SCI patients. First, FA images were created by fitting a tensor model to the raw diffusion data using FSL’s Diffusion toolbox (FDT), and then brain-extracted using a brain extraction tool (BET) of FSL [27]. Participants’ FA data were aligned into a 1 × 1 × 1 mm^3^ MNI152 space (a normalized/averaged brain developed by the Montreal Neurological Institute) by using the nonlinear registration tool FNIRT [28,29], which uses a b-spline representation of the registration warp field [30]. Next, the mean FA image was then created and thinned to create a mean FA skeleton that represents the centers of all tracts common to the group. DTI and DBSI indices were projected onto this skeleton for statistical analyses. Nonparametric permutation tests were used for voxel-wise statistical analysis of the FA skeletons between health controls and SCI patients. The significance threshold for group differences was set at *p* < 0.05. Thresholding was corrected for multiple comparisons across voxels by using the threshold-free cluster-enhancement option of the tool RANDOMIZE in FSL [31].

## 3. Results

### 3.1. Patient Characteristics

The mean age at the time of imaging for patients was 41 years (range 15–66 and 39 for controls (range 19–68). Patient demographics are summarized in Table 1. Months since injury ranged from 1 month to 480 months. All patients with an ASIA A or B injury were at least six months out from their injury (average 103 months, range 11–480 months). We attempted to include a heterogeneous patient population reflecting various levels and severity of injury. Patients and controls were not directly age matched, but the final demographic distributions were similar in all measured parameters for all cohorts. The most common causes of injury were falls, followed by motor vehicle collisions.

### 3.2. DTI Results

As depicted in Figure 1 and Table 2, DTI indices (FA, AD, and RD) from ASIA-A/B patients are different from controls only in CP region. FA values were significantly lower at the level of CP. In contrast, RD and AD values were both elevated at the level of the CP as compared to normal controls.

### 3.3. DBSI Results

As demonstrated in Figure 2 and Table 3, DBSI showed decreased FA values in the CP from cervical ASIA D to ASIA A/B, but not as reach significance as with DTI. AD values of ASIA A/B patients were significantly lower in CP, while RD values not different, compared to controls. In contrast to DTI, edema/tissue loss can be accounted for with DBSI measures. As depicted in Figure 3, increased non-restricted fraction in the CP contributed to the elevations of both DTI AD and RD, which also explains the over-estimated DTI FA drop in CP.

### 3.4. TBSS Results

A subsequent confirmatory analysis was carried out with TBSS. Tract-based spatial statistics comparison of all SCI cohorts with healthy controls demonstrated no significant differences.

## 4. Discussion

While conventional MRI provides details about the degree of spinal cord compression, parenchymal edema, and evidence of hemorrhage after an acute SCI, it provides no information about preserved axonal integrity [32,33]. Increasingly, DTI has been proposed as a surrogate marker for injury in both CSM and SCI [18,34,35,36,37]. In chronic disease, we would expect axonal loss at the zone of injury but what has remained unclear is whether Wallerian degeneration extends rostrally above the cortical spinal tracts and into the brain. Previous work with DTI has shown that DTI metrics are confounded by increased cellularity and vasogenic edema in ongoing states of acute or ongoing compression such as CSM or acute SCI. DBSI has been proposed to address DTI limitations by resolving multiple-tensor water diffusion resulting from axon injury, demyelination, and inflammation [4]. In a chronic disease state, we hypothesized both DTI and DBSI would demonstrate axonal loss extending to the level of the CP and IC, resulting in decreased FA and AD values that correlate with disease severity. DTI identified changes of FA, AD, and RD for ASIA-A/B patients at the level of CP. However as increasingly reported in the literature, interpretations of DTI indices are not specific as pathologic complexity increases. DBSI results suggested axonal injury and edema/tissue loss in CP for ASIA-A/B patients that increased with injury severity. Indeed, Cervical ASIA A/B SCI patients had higher levels of axonal injury and edema/tissue loss as measured by DBSI at the level of the cerebral peduncle. DTI was able to detect differences in cervical ASIA A/B patients (FA, AD, and RD), but they were non-specific to pathologies. Increased water fraction indicated by DBSI non-restricted isotropic diffusion fraction in the cerebral peduncle explains the simultaneously increased DTI AD and DTI RD values. Our results further demonstrate the utility of DTI to detect disruption in axonal integrity in white matter, yet a clear shortcoming in differentiating true axonal injury from inflammation/tissue loss.

The whole-brain analysis using TBSS failed to detect any statistical difference on all DTI and DBSI indices. While this automated whole-brain analysis can reveal locations of effects globally, minor changes may be harder to detect by TBSS because voxels more distant from tract centers contribute less to the average value projected on the skeleton [38].

Although SCI does not result in injury to cortical neurons, it does produce a physiologic disconnection of the distal motor targets, and it disassociates the sensorimotor areas. The cortical reorganization that occurs after SCI is not well defined. Previous authors have suggested there is loss of grey and white matter volume in cervical SCI patients as compared to normal controls [39]. Using functional MRI (fMRI) and positron emission tomography (PET) imaging, multiple authors have also reported increased cortical and subcortical activation in patients with SCI [40,41,42,43]. By contrast, several authors have reported no change or decreased cortical activation in individuals with SCI [44,45,46,47]. The significance of fMRI and PET studies remains unclear and provides no direct information regarding preserved axonal integrity. What does seem abundantly clear is that cortical reorganization occurs following SCI, but how this affects one’s potential for subsequent functional recovery or response to adjunctive treatment remains unclear [48,49,50].

Recently, Kurpad SN et al. [50] described changes that occur in network connectivity following SCI using resting state fMRI. The authors reported reduced functional connectivity in the sensorimotor cortex from SCI patients as compared to normal controls, yet an increase in connectivity with the thalamus. These dynamic changes seem intuitive given the known mismatch in efferent and afferent axons that occurs after SCI and further supports previous fMRI and PET studies that neural plasticity does allow for a shift in cortical connectivity, but what still remains unclear is whether this leads to subsequent axonal loss.

Only recently has DTI been used to investigate the cortical and subcortical changes that are thought to occur after SCI. In patients with chronic SCIs (i.e., >24 months), several authors have reported DTI metrics of brain CSTs decrease as compared with normal controls [51,52]. While we did find increased RD, and decreased FA and AD at the level of the CPs, this difference resolved rostrally at the level of the IC. However the AD, a marker for axonal injury, is supposed to decrease, not increase. Due to the limitation of single tensor model of DTI, both AD and RD will be elevated when edema/tissue loss is present. In such case, DTI overestimates axonal injury and demyelination. This loss would be expected with Wallerian degeneration occurring above the level of SCI. What is not well-defined is how much degeneration occurs at the level of internal capsule and cortically. DBSI was developed to address multiple co-existing pathologies by modeling anisotropic diffusion components (fiber tracts) from isotropic ones (structures and pathologies surrounding fiber tracts). Thus, DBSI can quantitatively evaluate fiber integrity, as well as edema/tissue loss and inflammation.

In the current study, DBSI results demonstrated both axonal injury and edema/tissue loss occurred at the level of CP in cervical ASIA-A/B patient, but not in any other patients as compared to normal controls. Furthermore, no differences were observed in any of the groups for any of the metrics at the level of the internal capsule. While cortical reorganization has been described following traumatic SCI [50,53,54,55], our results may help explain why patients, even years after a complete SCI, respond to epidural stimulation [56,57,58,59] and why recovery was observed following nerve transfers for SCI [60,61]. The loss of somatosensory input may redirect cortical activity, but as suggested by our results, does not necessarily lead to axonal loss. These results are promising and have implications for future trials aimed at treating chronic SCI.

*Limitations*—There are several limitations to consider from the presented data. First is the low numbers of patients, which may potentially account for the lack of any significant differences across the SCI cohorts. Second is lack of additional connectivity data, ideally this data could be corrected with resting state fMRI to assess concurrent changes in the various states of SCI.

## 5. Conclusions

MRI and increasingly advanced sequences such as DTI and DBSI provide guidance in the management and diagnosis of CNS pathologies. DTI and DBSI have been proposed as non-invasive tools that measure both axonal injury and demyelination. In this small subset of patients, DTI and DBSI measures of axonal loss and demyelination did not correlate with patient functional status. These results are promising to future regenerative treatment strategies, demonstrating axons may be preserved rostrally at the site of injury. This preservation at a higher cortical level may provide some insight as to why some patients respond to more recent epidural stimulation even years out from injury.

## Figures and Tables

**Figure 1 brainsci-07-00021-f001:**
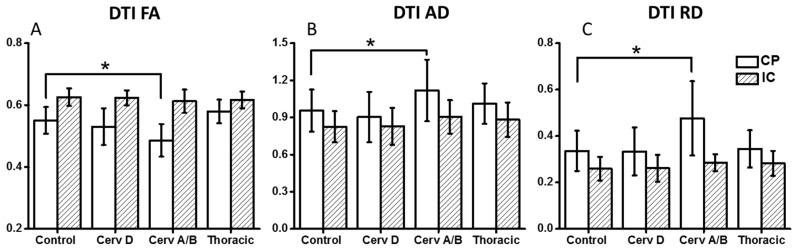
Compared to controls, DTI FA (**A**); AD (**B**) and RD (**C**) for Cervical ASIA-A/B demonstrated decreased FA and increased AD/RD in cerebral peduncle (CP) and internal capsule (IC). * *p* < 0.05. DTI—Diffusion Tensor Imaging, FA—Fractional Anisotropy, AD—Axial Diffusivity, RD—Radial Diffusivity, ASIA—American Spinal Injury Association scale.

**Figure 2 brainsci-07-00021-f002:**
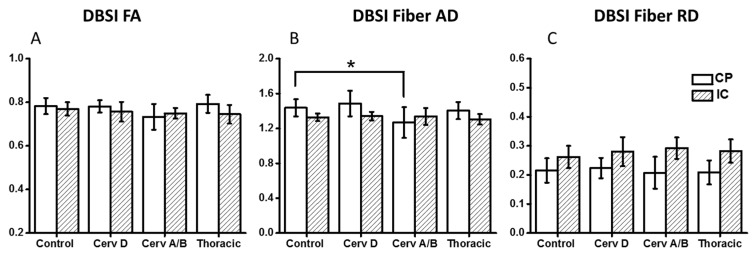
DBSI FA (**A**) and DBSI AD (**B**) and RD (**C**) values of Control, Cervical ASIA-A/B, Thoracic ASIA-A/B/C/D patients. Cervical ASIA-A/B patients had significantly lower DBSI AD values compared to controls and Cervical ASIA-D patients at the level of the cerebral peduncle. This difference was not observed higher up in the internal capsule. * *p* < 0.05. DBSI—Diffusion Basis Spectrum Imaging, FA—Fractional Anisotropy, AD—Axial Diffusivity, RD—Radial Diffusivity, ASIA—American Spinal Injury Association scale.

**Figure 3 brainsci-07-00021-f003:**
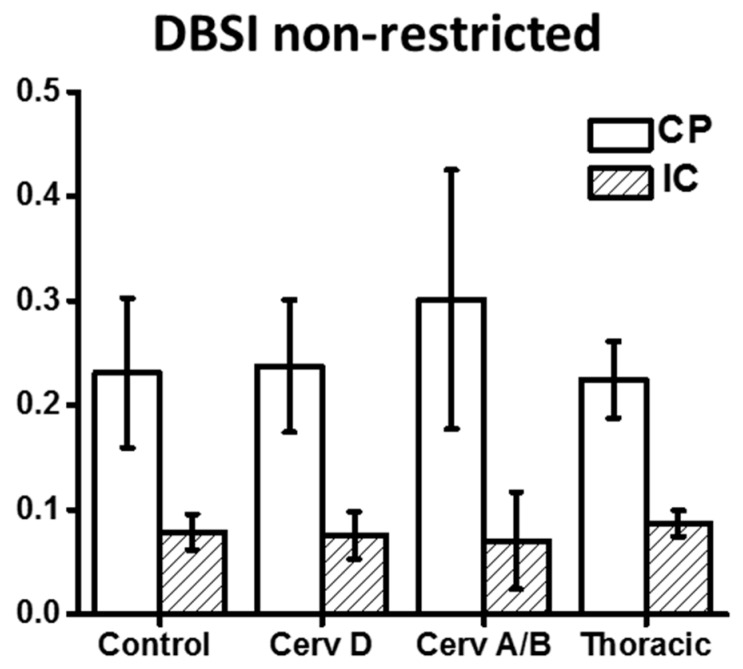
Compared to controls, DBSI derived non-restricted fraction at the level of the cerebral peduncle and internal capsule increase for cervical ASIA-A/B patients. DBSI—Diffusion Basis Spectrum Imaging, FA—Fractional Anisotropy, ADC—Apparent Diffusion Coefficient, AD—Axial Diffusivity, RD—Radial Diffusivity, ASIA—American Spinal Injury Association scale, CP—Cerebral Peduncle, IC—Internal Capsule.

**Table 1 brainsci-07-00021-t001:** Patient Demographics.

ID	Age	Race	Sex	Age at Injury	Months Since Injury	Mechanism of Injury	Severity	Initial ASIA	Initial Site of Injury	Isncsci Motor Max 100	Isncsci Sensory Max 224
1	46	AA	M	46	1	Fall	Incomplete	D	C5	96	224
2	15	AA	M	15	1	Fall	Incomplete	D	C6	90	224
3	41	AA	M	32	107	MVC	Complete	A	C4	5	28
4	45	C	F	44	16	MVC	Complete	A	T9	50	120
5	27	C	F	25	18	Fall	Incomplete	D	C6	77	219
6	40	C	M	21	233	Baseball	Complete	A	C4	10	32
7	66	AA	F	65	13	MVC	Complete	A	C6	30	88
8	54	C	F	50	40	Fall	Incomplete	D	C6	42	216
9	26	C	M	26	11	MVC	Complete	A	T5	50	84
10	56	C	M	56	2	Fall	Incomplete	D	C6	95	216
11	32	C	M	20	145	MVC	Incomplete	C	T4	90	156
12	48	C	M	47	16	MVC	Complete	A	C6	10	88
13	50	AA	M	50	3	Fall	Incomplete	D	C4	98	220
14	19	C	F	16	30	MVC	Incomplete	D	T12	70	184
15	36	C	M	35	19	Fall	Complete	A	T11	50	138
16	59	AA	M	59	1	Fall	Incomplete	D	C5	98	224
17	24	C	M	20	45	MVC	Complete	A	T9	50	176
18	68	C	M	28	480	MVC	Incomplete	B	C8	46	156
19	57	C	M	57	1	Hit by 500lb sign	Incomplete	D	T7/T8	92	208
20	53	C	M	52	3	Fall	Incomplete	D	C4	82	132
21	56	C	M	52	55	MVC	Complete	A	C7	20	16
22	36	C	M	36	3	Motorcycle	Complete	A	L1	53	160
23	71	C	M	71	2	Fall	Incomplete	D	C4/C5	90	212

ASIA—American Spinal Injury Association Scale, C—Caucasian, AA—African American, M—Male, F—Female, MVC—motor vehicle collision.

**Table 2 brainsci-07-00021-t002:** DTI Results–Brainstem and internal capsule.

Region	Group	FA	ADC	AD	RD
Cerebral peduncle (CP)	Control	0.55 ± 0.04	0.54 ± 0.11	0.96 ± 0.17	0.34 ± 0.09
	Cervical D	0.53 ± 0.06	0.52 ± 0.13	0.90 ± 0.20	0.33 ± 0.10
	Cervical A/B	0.48 ± 0.06	0.69 ± 0.19	1.12 ± 0.25	0.48 ± 0.16
	Thoracic A/B/C/D	0.57 ± 0.04	0.56 ± 0.11	1.01 ± 0.16	0.34 ± 0.08
Internal capsule (IC)	Control	0.63 ± 0.03	0.45 ± 0.07	0.83 ± 0.13	0.26 ± 0.05
	Cervical D	0.62 ± 0.02	0.46 ± 0.08	0.83 ± 0.15	0.26 ± 0.06
	Cervical A/B	0.61 ± 0.04	0.53 ± 0.06	0.90 ± 0.14	0.28 ± 0.04
	Thoracic A/B/C/D	0.62 ± 0.03	0.48 ± 0.08	0.88 ± 0.14	0.28 ± 0.05

DTI—Diffusion Tensor Imaging, FA—Fractional Anisotropy, ADC—Apparent Diffusion Coefficient, AD—Axial Diffusivity, RD—Radial Diffusivity, ASIA—American Spinal Injury Association scale.

**Table 3 brainsci-07-00021-t003:** DBSI Results—Brainstem and internal capsule.

Region	Group	Fiber Fraction	Fiber FA	Fiber AD	Fiber RD	Non-Restricted Fraction
Cerebral peduncle	Control	0.53 ± 0.07	0.78 ± 0.04	1.44 ± 0.10	0.22 ± 0.04	0.23 ± 0.07
	Cervical D	0.49 ± 0.06	0.78 ± 0.03	1.49 ± 0.15	0.22 ± 0.04	0.24 ± 0.06
	Cervical A/B	0.50 ± 0.10	0.73 ± 0.06	1.27 ± 0.18	0.21 ± 0.05	0.30 ± 0.12
	Thoracic A/B/C/D	0.57 ± 0.07	0.79 ± 0.04	1.40 ± 0.10	0.21 ± 0.04	0.22 ± 0.04
Internal capsule	Control	0.74 ± 0.07	0.77 ± 0.03	1.33 ± 0.04	0.26 ± 0.04	0.08 ± 0.02
	Cervical D	0.72 ± 0.08	0.76 ± 0.05	1.34 ± 0.05	0.28 ± 0.05	0.08 ± 0.02
	Cervical A/B	0.77 ± 0.06	0.75 ± 0.02	1.34 ± 0.10	0.29 ± 0.04	0.07 ± 0.05
	Thoracic A/B/C/D	0.74 ± 0.07	0.75 ± 0.04	1.31 ± 0.06	0.28 ± 0.04	0.09 ± 0.01

DBSI—Diffusion Basis Spectrum Imaging, FA—Fractional Anisotropy, AD—Axial Diffusivity, RD—Radial Diffusivity.

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
