# Peer review of "Diffusion Assessment of Cortical Changes, Induced by Traumatic Spinal Cord Injury"

_brainsci, 2017, doi:10.3390/brainsci7020021_

Round 1

Reviewer 1 Report

This paper describes microstructural changes in the CP by means of ROI analysis. The methods used are sound however, the sample size is small and in fact the authors only revealed changes in the cervical AIS A population (5 individuals). The DTI findings are a replication of previous findings reported in the literature and the DBSI results add little more information.

In the introduction, the authors state the hypothesis that brain changes might be useful predictors of clinical function. The paper would benefit from additional analysis (e.g. cord atrophy measures, brain volume changes) in order to to understand the driving forces in the most severely affected individuals (i.e. lesion gradient, time since injury).

Minor comments:

The paper has many typos and in the content of the authors contribution section is wrong.

Why do the author not distinguish between complete and incomplete thoracic individuals?

Figure 4 is not necessary. The tables would benefit from more specific titles.

More information is required for the specifics of the DTI acquisitions. In particularly, which b values are used?

Author Response

Reviewer #1 : 

This paper describes microstructural changes in the CP by means of ROI analysis. The methods used are sound however, the sample size is small and in fact the authors only revealed changes in the cervical AIS A population (5 individuals). The DTI findings are a replication of previous findings reported in the literature and the DBSI results add little more information.

We agree with the reviewer, the current study does have a limited number of patients. We are currently collecting longitudinal data on this existing cohort and are actively working on recruiting additional patients into this study. Despite this limited number of patients we believe the data is important and adds to the existing literature. Even though both DTI and DBSI can detect changes in these patients, DBSI measures are sensitive to pathologies where concomitant edema or inflammation is present. As detailed in the manuscript based on previous DTI studies, DTI indices are sensitive but non-specific.  They can only be specific only when single pathologies are present. As previously reported, DTI axial diffusivity should decrease if axonal injury present, but becomes unpredictable if concurrent pathologies coexists, such as inflammation/edema/tissue loss. The Increase of radial diffusivity represents demyelination, yet inflammation/tissue loss can also falsely elevate RD levels. The mixture or overlap of these pathologies can cause uncertainty in the interpretation of DTI results. DBSI allows for accurate separation quantitative of the differing pathologies – axonal loss vs neuroinflammation. We found significantly increased edema/tissue loss that explains the overestimated DTI axial and radial diffusivity. We believe DBSI may have potential to more accurately delineate true axonal and myelin loss and more accurately longitudinally monitor patients.

In the introduction, the authors state the hypothesis that brain changes might be useful predictors of clinical function. The paper would benefit from additional analysis (e.g. cord atrophy measures, brain volume changes) in order to to understand the driving forces in the most severely affected individuals (i.e. lesion gradient, time since injury).

Thanks for the suggestion. We agree with the review that more analysis would be helpful, but is beyond the scope of the current study. In the current study, our focus was cortically based. The measurement of brain volume and white matter volumes didn’t suggest statistical difference among groups. There is no lesion can ben delineated in those patients.

Minor comments:

The paper has many typos and in the content of the authors contribution section is wrong.We have corrected the typos.

Why do the author not distinguish between complete and incomplete thoracic individuals?

Due to the limited number of thoracic patients and the similar metrics these were grouped together.

Figure 4 is not necessary.

Figure 4 has been removed from the manuscript.

The tables would benefit from more specific titles.

Titles have been added.

More information is required for the specifics of the DTI acquisitions. In particularly, which b values are used?

More detailed information about diffusion acquisition has been added. Diffusion vector table information was added as table 1. 

Reviewer 2 Report

The objective of this study was to investigate whether DTI/DBSI changes that extend to level of the cerebral peduncle and internal capsule following a SCI could be correlated with clinical function. The authors meet this objective. 

The hypothesis to use DTI and DBSI metrics of the brainstem and internal capsule  a predictor of clinical function in chronic SCI patients is of interest and would be of benefit for the field.

However, the results could be elaborated on and explained more clearly.

Much of the discussion focuses on previous results. More space should be dedicated to the results presented in the current paper. For the paper to be of interest to others in the field, the authors need to examine in more detail why the measures of axonal loss and demyelination did not correlate with the functional status and could discuss possible measures to overcome these shortcomings.

Author Response

Reviewer #2: 
The objective of this study was to investigate whether DTI/DBSI changes that extend to level of the cerebral peduncle and internal capsule following a SCI could be correlated with clinical function. The authors meet this objective. 

The hypothesis to use DTI and DBSI metrics of the brainstem and internal capsule  a predictor of clinical function in chronic SCI patients is of interest and would be of benefit for the field.

However, the results could be elaborated on and explained more clearly.

Much of the discussion focuses on previous results. More space should be dedicated to the results presented in the current paper. For the paper to be of interest to others in the field, the authors need to examine in more detail why the measures of axonal loss and demyelination did not correlate with the functional status and could discuss possible measures to overcome these shortcomings.

The DBSI measure of axon loss and demyelination suggested more axonal loss and demyelination in severe group, but the changes were not significantly different by statistics. There may be two reasons for this. One is the limited number of patients, which weaken the power of statistics. We are recruiting more patients for this study. The second is the higher image quality required by DBSI. 15 minutes of diffusion MRI is longer than regular DTI, thus the control of motion, especially for severe patients become more difficult. We currently use post-processing to correct such motion. But this can be overcame by shortening imaging time and improving patient handling. Newer scanner hardware and sequences in our facility have tested, half of the acquisition time has been achieved. 

Round 2

Reviewer 1 Report

This revision has improved the paper slightly. Still the low number of patients and the fact that no significant results was obtained with the DBSI method at the whole group levels prevents a sound inference on whether this methods adds to novelty. The English is still rather poor. The cohort selection is not well balanced as the authors state themselves in the reply letter. In order to support their conclusion further analysis on a bigger cohort would be required before publishing this report. Perhaps the inclusion of longitudinal data would provide insights. The discussion on the functional relevance of their findings could improve further.

Minor:

Abstract is too long and should be more concise.

The diffusion vector Table is not helpful. This could be placed into the appendix.

Figure 4 is still not removed.

The reference list has hyperlinks. Pls remove.

Author Response

December 12th, 2016

Leia Lv

Assistant Editor

Brain Sciences

Dear Dr. Lv,

Thank you for sending us the reviewers’ comments regarding our manuscript, “Diffusion Assessment of Cortical Changes, Induced by Traumatic Spinal Cord Injury” (ISSN 2076-3425). We are pleased that the reviewers found the manuscript to be of interest. In response to their comments, we have made the following changes to the manuscript and below is a detailed point by point response.

Response to reviewers’ comments:

Reviewer #1 : This revision has improved the paper slightly. Still the low number of patients and the fact that no significant results was obtained with the DBSI method at the whole group levels prevents a sound inference on whether this methods adds to novelty. The English is still rather poor. The cohort selection is not well balanced as the authors state themselves in the reply letter. In order to support their conclusion further analysis on a bigger cohort would be required before publishing this report. Perhaps the inclusion of longitudinal data would provide insights. The discussion on the functional relevance of their findings could improve further. – We agree with the reviewer, ideally we would have many patients stratified on degree and level of injury with years of longitudinal data. We respectfully disagree that sort of data is necessary for this publication. As discussed in the results and discussion section we described our preliminary work which suggest 1)As previously described in the literature DBSI may provide a more accurate assessment of true white matter pathology inclusive of the inflammatory component 2) Demonstrate the preservation of normal cortical axon volume even in the setting of long standing severe cervical and thoracic SCI. We hope the editor and reviewer will consider the merits of the paper and we believe this will further contribute to the existing literature.

Minor:

Abstract is too long and should be more concise. We have shortened the abstract.

The diffusion vector Table is not helpful. This could be placed into the appendix. We have moved this to the appendix.

Figure 4 is still not removed. – the original figure 4 of the TBBS has been removed if the reviewer is referring to a different figure than the TBBS we would be happy to accommodate that request.

The reference list has hyperlinks. Pls remove. – we have removed.